# An Overview of Mechanical Properties of Diamond-like Phases under Tension

**Julia A. Baimova** [1,2]

1 Institute for Metals Superplasticity Problems, Russian Academy of Sciences, 450001 Ufa, Russia; julia.a.baimova@gmail.com
2 The World-Class Advanced Digital Technologies Research Center, Peter the Great St. Petersburg Polytechnic University, 195251 St. Petersburg, Russia

**Abstract:** Diamond-like phases are materials with crystal lattices very similar to diamond. Recent results suggest that diamond-like phases are superhard and superstrong materials that can be used for tribological applications or as protective coatings. In this work, 14 stable diamond-like phases based on fullerenes, carbon nanotubes, and graphene layers are studied via molecular dynamics simulation. The compliance constants, Young's modulus, and Poisson's ratio were calculated. Deformation behavior under tension is analyzed based on two deformation modes—bond rotation and bond elongation. The results show that some of the considered phases possess very high Young's modulus ($E \geq 1$) TPa, even higher than that of diamond. Both Young's modulus and Poisson's ratio exhibit mechanical anisotropy. Half of the studied phases are partial auxetics possessing negative Poisson's ratio with a minimum value of $-0.8$. The obtained critical values of applied tensile strain confirmed that diamond-like phases are high-strength structures with a promising application prospect. Interestingly, the critical limit is not a fracture but a phase transformation to the short-ordered crystal lattice. Overall, our results suggest that diamond-like phases have extraordinary mechanical properties, making them good materials for protective coatings.

**Keywords:** graphene; diamond-like phases; elastic constants; molecular dynamics; mechanical properties

## 1. Introduction

A rich variety of new carbon nanostructures have been reported, such as diamond, diamond-like phases (DLP), nanodiamonds, cubic carbon, and diamond particles to name a few. Such structures possess unique optical, mechanical, and physical properties [1–6]. Among them are cubic carbon BC12 [3], which can be used as a semiconductor; body-centered cubic BC8 carbon [7,8]; simple cubic carbon phase-termed SC24 [9]; and an O16 superhard phase [10]. More recently, a rhombohedral carbon phase R16 carbon [11] has been reported and indicated to be the derivation of the milled fullerene soot [12]. To note, BC8, O16, and BC12 carbon comprises entirely a diamond-like six-membered ring bonding structure, while other $sp^3$ hybridized carbon structures contain mixed bonding configurations, such as the $(m + n + k)$-membered rings, where $m, n, k = \{3:8\}$. A complete description of structural characteristics and basic properties of carbon allotropes can be found in the SACADA database [13]. Diamond-like phases possess characteristics similar to that of diamond, like super-hardness and super-strength, and the possibility to be semiconductors with a direct band gap [3–5].

Such new carbon phases can be obtained under extreme conditions, such as high pressure or shock compression, because it can induce new bond modifications [3,4,7,8,14–18]. Another way to synthesize carbon phases is the heating of carbon soot or shock compression of polycrystalline graphite [19,20]. Some DLPs, for example, BC8, can be derived from cubic diamond under high pressure [8]. Two structural forms have been obtained from cold-compressed graphite: a monoclinic M carbon [4] and an orthorhombic W carbon [21].

Diamond-like carbon materials possess numerous excellent properties like high hardness, low wear, low friction coefficient, chemical inertness, and biocompatibility [22–24]. The carbon phase obtained in [5] is superhard as evidenced by the broadening of ruby fluorescence lines and its ability to indent diamond anvils. DLPs possess Vickers hardness $H_V$ that characterizes the resistance of a crystal to plastic deformation under local force action from 49.4 to 90.0 GPa [25]. The maximum hardness demonstrates DLPs with the highest densities. Hardening of such DLPs via structure modification can be compared to the model of fcc hard-sphere crystals presented in [26]. New carbon phase O16 possesses a large Bulk modulus (435 GPa) and Vickers hardness (93 GPa). Often, DLPs exhibit chemical stability, low friction, high hardness, and high strength close to or even exceeding that of the perfect cubic diamond structure [27]. Therefore, DLP films have been widely used as protective coatings for improving friction and wear properties of different surfaces [28,29]. Moreover, DLPs are very promising as protective coatings for nano-electromechanical systems and modern electronic devices [30]. Thus, analyzing the mechanical properties, tribological, and friction behavior is important for enhancing the functional design of new nano-electromechanical systems.

An important technique in diamond synthesis is inducing a phase transition of molecular crystals made of carbon polymorphs. High compression and temperatures provide the compression of the structure, breaking of bindings, and formation of new bonds. One of the important classes of DLPs are structures obtained based on fullerenes or fullerene-like molecules, carbon nanotubes (CNT), and graphene layers [23,31–36]. Especially, $C_{60}$ fullerene is the most widely used raw material in diamond synthesis [37–39].

Previous studies of carbon structures have shown that some of them can possess a negative Poisson ratio at various conditions, like tension or the presence of defects [40]. Structures that possess negative Poisson's ratio are called auxetics; however, the term "auxetic" was introduced only in 1991 in [41]. One of the first research works on the mechanical model of the structures with negative Poisson's ratio was presented in [42]. Since that time, the idea of auxetic materials has become very popular. Lakes manufactured the first auxetic foam in 1987 [43]. Further, the first microscopic model of the hard cyclic hexamers with a thermodynamically stable isotropic phase exhibiting negative Poisson's ratio was simulated by the Monte Carlo method in [44] and then its certain generalization (soft cyclic hexamers) was analytically solved in the static limit at zero temperature in [45]. To date, there is a very clear classification of auxetics: complete auxetic with negative Poisson's ratio in all directions, partial auxetics for systems with negative Poisson's ratio in some directions, and non-auxetic for the rest of materials [46]. A wide variety of auxetic structures can be found: composite auxetics was described in a pioneering work [47], and cubic auxetics we shown in [48]. A well-known rotational planar model to explain auxeticity was proposed in [49]. Computer simulations of an important molecular auxetic were described in [50]. Recent theoretical or simulation works concerning some fundamental auxetic structures and mechanisms at the micro level allow us to understand the nature of auxeticity [51,52].

Syntheses of such DLPs is a complex technological process, and their optimization requires detailed knowledge of the corresponding atomistic mechanisms. Various methods have been used to study the fabrication processes in various carbon systems. Advances in the development of the nanostructure are determined by the development of new manufacturing technologies, as well as methods for complex property analysis (numerical simulations and analytical methods). For example, molecular dynamics (MD) is the method that allows for describing the motion of atoms or particles using the method of classical mechanics, both to represent the valence and van der Waals forces and to consider systems with many thousands of atoms in nanosecond time intervals. In carbon nanoscience, MD was successfully used for the investigation of secondary structures [53], thermal conductivity [53–55], deformation behavior and mechanical properties [55–57], etc. The development of the reactive interatomic potentials for carbon [58–63] adopted molecular dynamics (MD) as one of the most used tools for numerical calculations in this field. Existing MD simulations allow for analyzing the phase transition during synthesis

or deformation of carbon phases [39,64], their stability [65], and their properties in a wide variety of external effects [66,67]. MD simulations can be utilized as a phenomenological method to understand the deformation behavior in materials at the atomistic level. Simulations at the atomistic level can provide both physical and mechanical properties and detailed information about structural changes.

In the present work, the mechanical properties of diamond-like phases were analyzed using molecular dynamics simulation. Three classes of diamond-like phases were considered: fulleranes composed of fullerenes, tubulanes consisting of carbon nanotubes, and DLPs based on graphene layers. The compliance constants, Young's modulus, and Poisson's ratio are analyzed. Deformation behavior is analyzed based on the understanding of two deformation modes—rotation and elongation of bonds.

## 2. Simulation Details

The calculations were carried out using the MD simulation as implemented in the LAMMPS simulation package [68–70], with AIREBO interatomic potential [60]. The AIREBO parameterization is based on a wide range of experimental data on the properties of carbon structures and hydrocarbons, which allows for a successive use of this potential for a wide variety of issues [30,67,71–73]. However, for example, to study different paths for the phase transformations, it is better to use the well-known ReaxFF potential [35,74,75].

Very different interatomic potentials can be used to analyze the properties of carbon nanostructures. Among them are REBO or Brenner [59], AIREBO [60], Tersoff [76], ReaxFF [61], LCBOP [62], and the standard set of interatomic potentials [63], to name a few. All of them take into account the breaking and the formation of bonds while having their pros and cons; however, most include a bond-order-dependent part, a dihedral part, and van der Waals interactions. ReaxFF, for example, also includes Coulomb forces to describe non-bonded interactions between atoms. Each of the named interatomic potentials was successfully used for the investigation of different aspects of carbon nanostructures' behavior and each has its limitations, and sometimes results obtained by different potentials are significantly different. Owing to the approximate nature of empirical potentials, and the compromises involved in their construction, it is necessary to identify their limitations to justify their use [71,77–82]. For AIREBO, more $sp^2$ or $sp^3$ carbon nanostructures are formed in comparison with the ReaxFF; however, AIREBO can be used for the study of melting [71] and deformation behavior under strain [72,73] for carbon nanostructures. For example, a comparison of AIREBO, modified AIREBO-M, and LCBOP showed that all the potentials better reproduce the melting of graphite [71], and the melting temperature is close to the experimental one. AIREBO and some other empirical potentials are mostly able to reproduce the results of DFT calculations, but some notable failures are not necessarily fatal. For example, the AIREBO potential performs well when the activation energies for the formation and removal of a Stone–Wales defect and different types of vacancies are considered, giving a result that is fairly close to DFT [82]. AIREBO has been shown to well represent the binding energy and elastic properties of carbon materials but usually suffers from a non-physical high tensile stress which originated from the fixed switching function [78]. To overcome this problematic issue, the cut-off distance is usually extended far from the original value of 1.7 Å to 1.9–2.0 Å. In the present work, the adaptive cutoff parameter of the potential has been set to 2.0 Å to better describe the near-fracture regime. It was also shown that the AIREBO potential can effectively reproduce mechanical properties, even better than other empirical potentials [83].

Figure 1 shows the initial structural element, an enlarged part of the simulation cell, and the whole structure in a perspective. All three classes of DLPs are presented: fulleranes composed of fullerenes (a), tubulanes composed of carbon nanotubes (b), and DLPs based on graphene layers (c). According to [31], different fulleranes can be generated from the same fullerene-like molecules because of two ways of linking: DLPs termed by A are generated by the linking of two fullerenes with covalent bonding, while DLPs termed by B are generated by the uniting of carbon atoms on two fullerenes. The notation C is used

for fulleranes, T is used for tubulanes, and L is used for DLPs based on graphene layers. Number from one to nine after letters C, T, or L simply denote the index number.

The main parameters of DLPs are presented in Tables 1–3, where $L_1$–$L_4$ are bond lengths for each atom, $a$ is the lattice parameter in $x$ and $y$ directions, $c$ is the lattice parameter in the $z$ direction, and $\rho$ is the density of the structure. The size of the computational cell depends on the type of single structural element and the size of the primitive cell. The computational cell contains $8 \times 8 \times 8$ periodic cells along the $x$, $y$, and $z$ directions with the periodic boundary conditions applied. The final size of the simulation cell is presented in Tables 1–3. Due to the periodicity of the boundary conditions, the considered DLP can be considered as an infinite crystal; however, in real experiments, the thickness can considerably affect the mechanical properties of DLPs [84–86].

**Table 1.** Structural characteristics of tubulanes.

| Structures | Size, Å $L_x \times L_y \times L_z$ | $L_1$, Å | $L_2$, Å | $L_3$, Å | $L_4$, Å | $a$, Å | $c$, Å | $\rho$, g/cm$^3$ |
|---|---|---|---|---|---|---|---|---|
| TA1 | $25 \times 25 \times 22$ | 1.4921 | 1.5638 | 1.5638 | 1.5723 | 6.461 | 2.577 | 2.966 |
| TA3 | $25 \times 25 \times 22$ | 1.5857 | 1.5015 | 1.5625 | 1.5625 | 3.558 | 4.314 | 2.921 |
| TA5 | $30 \times 25 \times 20$ | 1.4846 | 1.5539 | 1.5612 | 1.5832 | 6.917 | 4.406 | 3.027 |
| TA6 | $26 \times 28 \times 30$ | 1.4869 | 1.5410 | 1.5928 | 1.5928 | 7.007 | 4.165 | 3.122 |
| TA8 | $20 \times 20 \times 22$ | 1.4830 | 1.5840 | 1.5587 | 1.6879 | 10.582 | 2.511 | 2.949 |
| TB | $17 \times 17 \times 19$ | 1.4984 | 1.4984 | 1.5152 | 1.5152 | 4.421 | 2.530 | 2.794 |

**Table 2.** Structural characteristics of fulleranes.

| Structure | $L_x \times L_y \times L_z$, Å | $L_1$, Å | $L_2$, Å | $L_3$, Å | $L_4$, Å | $a$, Å | $\rho$, g/cm$^3$ |
|---|---|---|---|---|---|---|---|
| CA2 | $17 \times 17 \times 17$ | 1.4411 | 1.5431 | 1.5437 | 1.5726 | 5.037 | 2.203 |
| CA3 | $20 \times 20 \times 20$ | 1.4464 | 1.5702 | 1.5702 | 1.5702 | 4.811 | 2.867 |
| CA7 | $15 \times 15 \times 15$ | 1.5014 | 1.5014 | 1.5665 | 1.5665 | 7.380 | 2.382 |
| CA8 | $20 \times 20 \times 20$ | 1.5047 | 1.5998 | 1.4966 | 1.5833 | 8.967 | 2.656 |
| CA9 | $21 \times 32 \times 25$ | 1.5389 | 1.4830 | 1.6139 | 1.5107 | 12.244 | 2.086 |
| CB | $20 \times 20 \times 20$ | 1.4990 | 1.5404 | 1.4919 | 1.4919 | 9.396 | 2.308 |

**Table 3.** Structural characteristics of DLP based on graphene.

| Structure | $L_x \times L_y \times L_z$, Å | $L_1$, Å | $L_2$, Å | $L_3$, Å | $L_4$, Å | $a$, Å | $c$, Å | $\rho$, g/cm$^3$ |
|---|---|---|---|---|---|---|---|---|
| LA3 | $16 \times 16 \times 20$ | 1.5225 | 1.5225 | 1.5672 | 1.5672 | 4.348 | 2.516 | 3.356 |
| LA6 | $15 \times 20 \times 20$ | 1.4837 | 1.5441 | 1.5441 | 1.5957 | 4.821 | 2.578 | 3.071 |

The building element for fulleranes are fullerene-like molecules $C_6$ (for CA2), $C_8$ (for CA3 or supercubane), $C_{24}$ (for CA9), and $C_{48}$ (for CA7, CA8, CB). Fullerene-like molecules and corresponding CA phases are presented in Figure 1a. The building element for tubulanes are CNTs (for TA1), CNT(2,0) (for TA3), CNT(4,0) (for TA5), CNT(4,0) (for TA6), CNT(3,3) (for TA8), and CNT(3,3) (for TB). Carbon nanotubes and the corresponding TA phase are presented in Figure 1b. Figure 1c presents the example of DLPs based on a graphene layer. Despite in [23,31,33,34,36] a greater number of structures being presented, MD simulation shows that there are only 14 stable DLPs of this structural type possessing different anisotropy: cubic CA3, CA7, CA8, CA9, and CB; tetragonal TA1, TA3, TA5, TA6, and LA3; hexagonal CA2 and TB; trigonal TA8; and orthorhombic LA6.

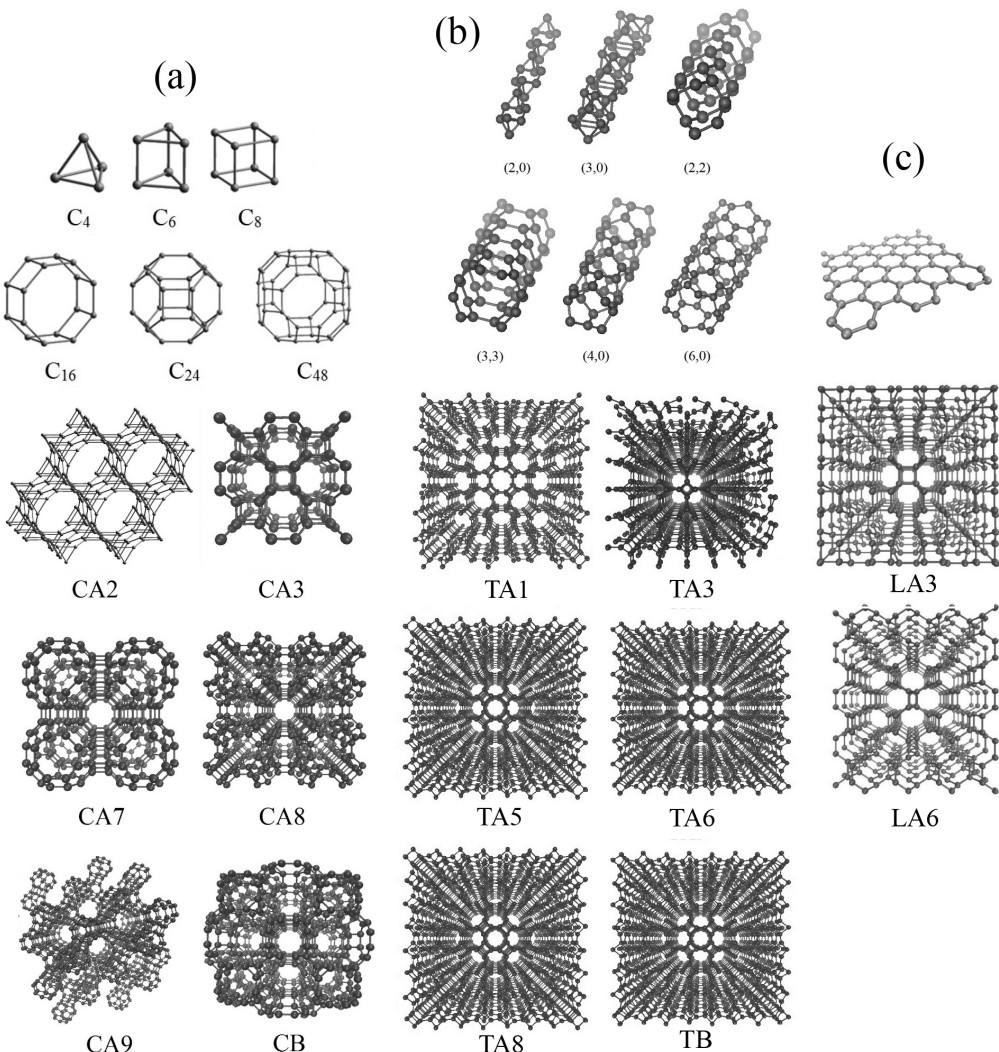

**Figure 1.** Single element of DLP; part of the simulation cell in a perspective view. (**a**) Fullerene-like molecules and fulleranes. (**b**) CNTs and tubulanes. (**c**) Graphene and DLPs' LA3 and LA6.

All the simulations are performed under the periodic boundary conditions aligned along all three directions. In our MD simulations, the velocity-Verlet algorithm is used to calculate the motion of each atom with a time step of 0.5 fs. After DLPs are created by the homemade program using parameters presented in Tables 1–3, all structures were relaxed. To reach the equilibrium state for DLPs, the energy of the system was minimized via the multiple correction of the atomic positions using the steepest descent method, then terminated if the variation in the energy or force was less than a given value. All the structures are relaxed until the system reaches its local or global minimum of the potential energy. The simulation run is terminated when one of the stopping criteria (energy or force) is satisfied. After several numerical experiments with different minimization parameters, stopping tolerance energy $10^{-24}$ and stopping tolerance force $10^{-26}$ eVÅ$^{-1}$ were chosen. Stopping tolerance energy is dimensionless and met when the energy change between successive iterations divided by the energy magnitude is less than or equal to the tolerance. Specified force tolerance is given in force units since it is the length of the global force vector for all atoms. For example, in the present case, the setting of tolerance force $10^{-26}$ means no $x$, $y$, and $z$ component of the force on any atom will be larger than $10^{-26}$ eVÅ$^{-1}$ after minimization.

To calculate stiffness constants, a strain of up to 1–2% was applied to the simulation cell in an elastic regime. All the stress–strain components were calculated during the

simulation as implemented in the LAMMPS package. The imposed stress is controlled using a Nose-Hoover pressure barostat. All the calculations were performed at 0 K. Tensile or shear strain is applied such that the tensile ($\varepsilon_{xx}$ and $\varepsilon_{yy}$) or shear strain ($\varepsilon_{xy}$) components do not exceed 1%.

Stiffness coefficients for any anisotropy can be calculated using Hook's law.

The imposed stress is controlled using a Nose-Hoover pressure barostat. To calculate $s_{11}$ and $s_{12}$, uniaxial stress $\sigma_{xx}$ is applied and two strain components are measured $\varepsilon_{xx}, \varepsilon_{yy}$; to calculate $s_{13}$, $s_{23}$ and $s_{33}$; uniaxial stress $\sigma_{zz}$ is applied and three strain components are measured $\varepsilon_{xx}, \varepsilon_{yy}$ and $\varepsilon_{zz}$; and to calculate $s_{22}$, uniaxial stress $\sigma_{yy}$ is applied and strain component $\varepsilon_{yy}$ is measured. To calculate $s_{44}$, $s_{55}$ and $s_{66}$, shear stress $\sigma_{yz}$, $\sigma_{xz}$ or $\sigma_{xy}$ is applied correspondingly to obtain strain components $\varepsilon_{yz}, \varepsilon_{xz}$ and $\varepsilon_{xy}$. For the cases when uniaxial strain is applied, the material is allowed to change the dimensions along the perpendicular direction. Tensile deformation were performed at 0 K with the temperature control via the Nose-Hoover thermostat.

The elastic properties of orthorhombic crystals are characterized by nine independent compliance coefficients: $s_{11}$, $s_{22}$, $s_{33}$, $s_{44}$, $s_{55}$, $s_{66}$, $s_{12}$, $s_{13}$, $s_{23}$. Based on Hooke's law for orthorhombic anisotropy, the compliance coefficients are calculated as follows

$$s_{11} = \frac{\varepsilon_{xx}}{\sigma_{xx}}, \quad s_{12} = \frac{\varepsilon_{yy}}{\sigma_{xx}}, \quad s_{13} = \frac{\varepsilon_{xx}}{\sigma_{zz}}, \quad s_{23} = \frac{\varepsilon_{yy}}{\sigma_{zz}}, \quad s_{33} = \frac{\varepsilon_{zz}}{\sigma_{zz}}. \tag{1}$$

$$s_{22} = \frac{\varepsilon_{yy}}{\sigma_{yy}}, \quad s_{44} = \frac{\varepsilon_{yz}}{\sigma_{yz}}, \quad s_{55} = \frac{\varepsilon_{xz}}{\sigma_{xz}}, \quad s_{66} = \frac{\varepsilon_{xy}}{\sigma_{xy}}. \tag{2}$$

Due to the positive definiteness of the elastic energy, there are necessary and sufficient elastic stability conditions on the compliance coefficients

$$\begin{gathered} s_{11} > 0, \quad s_{22} > 0, \quad s_{33} > 0, \quad s_{11}s_{22} > s_{12}^2, \\ s_{11}s_{22}s_{33} + 2s_{12}s_{13}s_{23} - s_{11}s_{23}^2 - s_{22}s_{13}^2 - s_{33}s_{12}^2 > 0, \\ s_{44} > 0, \quad s_{55} > 0, \quad s_{66} > 0. \end{gathered} \tag{3}$$

The above formula can be used to perform tetragonal anisotropy under the additional conditions: $s_{11} = s_{22}$, $s_{13} = s_{23}$, $s_{44} = s_{55}$.

Such elastic constants as Young's modulus and Poisson's ratio can be found from stiffness constants [72,73,87,88]. For anisotropic structures, Young's modulus and Poisson's ratio depend on the tensile direction with respect to the crystallographic axis.

To study the deformation behavior of DLPs under hydrostatic tension, tensile deformation was applied in such a way as to preserve the simulation cell volume. The computational cell is deformed in the hydrostatic regime ($\varepsilon_x = \varepsilon_y = \varepsilon_z = \varepsilon$), where $\varepsilon$ is the parameter that monotonically increases (or decreases) with the strain rate 0.005 ps$^{-1}$. Numerical calculations with the strain rate four and eight times smaller do not affect the results. Hydrostatic pressure and other system parameters are defined during deformation as implemented in LAMMPS. The time step of the simulations was 0.2 fs. The procedure of relaxation and stretching was repeated for all the structures to evaluate their mechanical properties. Tensile deformation was performed at 0 K with the temperature control via the Nose-Hoover thermostat.

## 3. Results and Discussion

Table 4 presents the compliance constants s$_{ij}$ of all the considered DLPs. As can be seen, there is a great difference in values depending on the DLP morphology. The elastic constants can be employed to examine the mechanical stability of DLPs including all phases. On the basis of the Born stability criteria [89], it was found that the stability criteria is satisfied for all DLPs, demonstrating the mechanical stability.

**Table 4.** Compliance constants $s_{ij}$ (in TPa$^{-1}$). The measurement (statistical) errors are also presented.

| DLP | $s_{11}$ | $s_{12}$ | $s_{13}$ | $s_{14}$ | $s_{22}$ | $s_{23}$ | $s_{33}$ | $s_{44}$ | $s_{55}$ | $s_{66}$ |
|---|---|---|---|---|---|---|---|---|---|---|
| | | | | | Cubic | | | | | |
| CA3 | 1.87 ± 0.3 | −0.44 ± 0.11 | | | | | | 2.496 ± 0.09 | | |
| CA7 | 8.12 ± 0.1 | −3.82 ± 0.2 | | | | | | 3.64 ± 0.4 | | |
| CA8 | 1.67 ± 0.3 | −0.299 ± 0.04 | | | | | | 5.91 ± 0.9 | | |
| CA9 | 3.73 ± 0.6 | −0.9 ± 0.08 | | | | | | 7.31 ± 0.9 | | |
| CB | 5.53 ± 0.4 | −2 ± 0.4 | | | | | | 10 ± 1.1 | | |
| | | | | | Tetragonal | | | | | |
| TA1 | 2.15 ± 0.04 | −0.18 ± 0.08 | −0.52 ± 0.04 | | | | 2.3 ± 0.2 | 7.9 ± 0.4 | | 5.51 ± 0.1 |
| TA3 | 1.9 ± 0.02 | −0.7 ± 0.08 | −0.52 ± 0.01 | | | | 2.58 ± 0.3 | 7.76 ± 0.3 | | 4.96 ± 0.2 |
| TA5 | 1.55 ± 0.2 | −0.5 ± 0.09 | −0.3 ± 0.09 | | | | 1.14 ± 0.14 | 5.6 ± 0.1 | | 2.16 ± 0.3 |
| TA6 | 1.47 ± 0.11 | −0.01 ± 0.002 | −0.13 ± 0.01 | | | | 0.82 ± 0.04 | 4.73 ± 0.4 | | 2.77 ± 0.9 |
| LA3 | 2 ± 0.1 | −2 ± 0.4 | −0.04 ± 0.001 | | | | 8.15 ± 0.4 | 11.5 ± 0.9 | | 3.44 ± 0.13 |
| | | | | | Trigonal | | | | | |
| TA8 | 1.98 ± 0.11 | −0.9 ± 0.04 | −0.06 ± 0.002 | −0.005 ± 0.001 | | | 0.96 ± 0.01 | 3.88 ± 0.18 | | |
| | | | | | Hexagonal | | | | | |
| CA2 | 2.51 ± 0.18 | 0.09 ± 0.004 | −0.36 ± 0.01 | | | | 1.92 ± 0.14 | 18.4 ± 0.3 | | |
| TB | 1.65 ± 0.18 | −0.16 ± 0.008 | −0.29 ± 0.007 | | | | 0.99 ± 0.03 | 10.17 ± 0.8 | | |
| | | | | | Orthorhombic | | | | | |
| LA6 | 2.45 ± 0.03 | −1.4 ± 0.1 | −0.3 ± 0.01 | | 2.2 ± 0.1 | −0.13 ± 0.06 | 1.2 ± 0.2 | 2.1 ± 0.4 | 2.3 ± 0.2 | 2.5 ± 0.3 |

From the stiffness constants listed in Table 5, we reveal that only CA7 of all DLPs shows the similar mechanical properties in both $x$ and $y$ directions ($c_{11}$ is almost equal to $c_{12}$). However, a strongly different mechanical behavior is also observed for other DLPs, whose $c_{11}$ is much larger than $c_{12}$. Such DLPs, such as TA6, TA5, TA3, CA7, and LA6, demonstrate the highest values of $c_{11}$ close to that of diamond. DLP TA6 demonstrates exceptional strength, with $c_{11}$ almost two times higher than for diamond. Phases CA2, CA9, and CB are the weakest among the considered DLPs.

**Table 5.** Stiffness constants $c_{ij}$ (in GPa). The measurement (statistical) errors are also presented.

| DLP | $c_{11}$ | $c_{12}$ | $c_{13}$ | $c_{14}$ | $c_{22}$ | $c_{23}$ | $c_{33}$ | $c_{44}$ | $c_{55}$ | $c_{66}$ |
|---|---|---|---|---|---|---|---|---|---|---|
| | | | | | Cubic | | | | | |
| Diam. | 1076 ± 120 | 125 ± 9 | - | - | - | - | - | 563 ± 4 | - | |
| CA3 | 625 ± 10 | 192 ± 12 | - | - | - | - | - | 401 ± 9 | - | |
| CA7 | 750 ± 18 | 667 ± 8 | - | - | - | - | - | 275 ± 4 | - | |
| CA8 | 650 ± 22 | 142 ± 4 | - | - | - | - | - | 169 ± 4 | - | |
| CA9 | 316 ± 11 | 101 ± 8 | - | - | - | - | - | 137 ± 5 | - | |
| CB | 306 ± 10 | 174 ± 4 | - | - | - | - | - | 100 ± 3 | - | |
| | | | | | Tetragonal | | | | | |
| TA1 | 652 ± 21 | 31 ± 8 | 196 ± 4 | - | - | - | 461 ± 8 | 130 ± 4 | - | 182 ± 4 |
| TA3 | 706 ± 21 | 294 ± 8 | 165 ± 11 | - | - | - | 463 ± 9 | 129 ± 5 | - | 201 ± 4 |
| TA5 | 820 ± 35 | 350 ± 9 | 226 ± 9 | - | - | - | 989 ± 14 | 179 ± 1 | - | 463 ± 3 |
| TA6 | 1854 ± 128 | 55 ± 2 | −8.59 ± 0.6 | - | - | - | 1214 ± 24 | 442 ± 12 | - | 221 ± 9 |
| LA3 | 626 ± 20 | 79 ± 2 | 40 ± 1 | - | - | - | 1232 ± 44 | 87 ± 1.4 | - | 290 ± 6 |
| | | | | | Trigonal | | | | | |
| TA8 | 657 ± 12 | 316 ± 4 | 65 ± 0.4 | 0.5 ± 0.09 | - | - | 1051 ± 54 | 257 ± 18 | - | - |
| | | | | | Hexagonal | | | | | |
| CA2 | 413 ± 18 | −1.6 ± 0.4 | 87 ± 1 | - | - | - | 555 ± 14 | 161 ± 3 | - | - |
| TB | 600 ± 18 | −30.5 ± 0.8 | 155 ± 5 | - | - | - | 1067 ± 38 | 3614 ± 141 | - | - |
| | | | | | Orthorhombic | | | | | |
| LA6 | 720 ± 33 | 473 ± 1 | 271 ± 4 | - | 761 ± 34 | 229 ± 9 | 933 ± 94 | 454 ± 14 | 423 ± 22 | 395 ± 23 |

Table 6 presents the maximal and minimal Young's modulus and Poisson's ratio for all considered DLPs. The details of the calculations are presented in [72,73,87,88]. For anisotropic structures, Young's modulus and Poisson's ratio, depending on the tensile direction with respect to the crystallographic axis, can considerably change over different directions [72,73,87,88]. Thus, here only maximal and minimal values are presented. The maximal values of Young's modulus, even bigger than for diamond [90,91], were found for TA6, TA8, TB, and LA3. Young's modulus of the diamond is 1144.6 GPa [90], and for the

CVD diamond, it is 1143 GPa. However, for DLP films, Young's modulus is almost two times lower: 500–530 GPa [92]; 541–875 GPa [93]. Very well-known DLPs such as ta-C:H, ta-C, and WWW-ta-C have a Young's modulus of 300, 757, and 829 GPa, respectively [90]. The temperature and density of DLPs can considerably affect the values of Young's modulus; for example, it changes from 150 to 370 GPa for different morphologies and conditions [66]. Figure 2 shows the graphical representation of maximal and minimal Young's modulus in comparison with diamond, DLP films, and other known DLPs.

**Table 6.** Young's modulus (in GPa), anisotropy ratio, and Poisson's ratio. The measurement (statistical) errors are also presented.

| DLP | $E_{max}$ | $E_{min}$ | $A_E$ | $\nu_{max}$ | $\nu_{min}$ |
|-----|-----------|-----------|-------|-------------|-------------|
| | | | Cubic | | |
| CA3 | 860 ± 23 | 535 ± 21 | 1.6 ± 0.3 | 0.33 ± 0.04 | −0.07 ± 0.01 |
| CA7 | 644 ± 13 | 123 ± 8 | 5.2 ± 0.3 | 1.14 ± 0.3 | −0.4 ± 0.11 |
| CA8 | 599 ± 19 | 430 ± 13 | 1.4 ± 0.1 | 0.37 ± 0.01 | 0.14 ± 0.04 |
| CA9 | 325 ± 7 | 268 ± 7 | 1.2 ± 0.01 | 0.28 ± 0.01 | 0.13 ± 0.04 |
| CB | 260 ± 3 | 181 ± 3 | 1.4 ± 0.07 | 0.47 ± 0.3 | 0.17 ± 0.06 |
| | | | Tetragonal | | |
| TA1 | 513 ± 11 | 299 ± 5 | 1.7 ± 0.09 | 0.58 ± 0.02 | 0.03 ± 0.009 |
| TA3 | 834 ± 33 | 380 ± 6 | 2.2 ± 0.11 | 0.42 ± 0.1 | 0.1 ± 0.004 |
| TA5 | 893 ± 19 | 573 ± 173 | 1.5 ± 0.05 | 0.57 ± 0.03 | 0.15 ± 0.04 |
| TA6 | 1852 ± 128 | 717 ± 19 | 2.6 ± 0.22 | 0.62 ± 0.09 | −0.01 ± 0.007 |
| LA3 | 1235 ± 99 | 293 ± 8 | 4.2 ± 0.3 | 0.51 ± 0.09 | −0.13 ± 0.001 |
| | | | Trigonal | | |
| TA8 | 1043 ± 87 | 504 ± 4 | 2.0 ± 0.4 | 0.48 ± 0.11 | 0.3 ± 0.04 |
| | | | Hexagonal | | |
| CA2 | 518 ± 18 | 393 ± 13 | 1.3 ± 0.19 | 0.27 ± 0.11 | −0.04 ± 0.01 |
| TB | 1730 ± 77 | 983 ± 4 | 1.6 ± 0.3 | 0.29 ± 0.1 | −0.8 ± 0.011 |
| | | | Orthorhombic | | |
| LA6 | 1039 ± 77 | 407 ± 24 | 2.5 ± 0.3 | 0.66 ± 0.04 | −0.14 ± 0.08 |

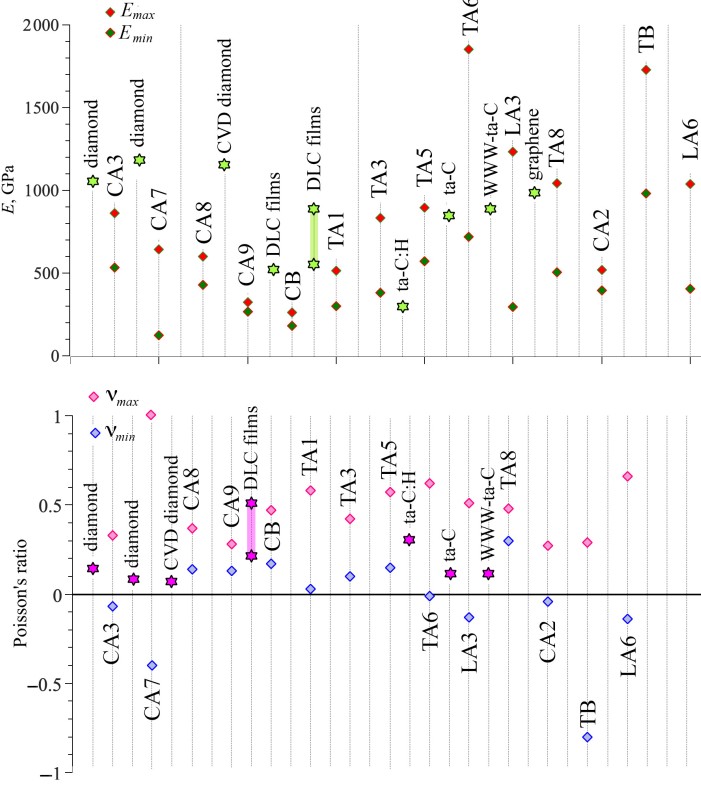

**Figure 2.** Young's modulus and Poisson's ratio for all studied DLPs (shown by rhombus) compared with the literature (shown by stars) [72,73,87,88,90–94].

The lowest Young's modulus is found for cubic C9 and CB phases. Phase LA3 possesses a considerable difference between maximal and minimal values of Young's modulus, about four times. It is very similar to graphite, which exhibits almost zero Young's modulus along the [001] direction and a maximal Young's modulus of 1000 GPa along the [010] direction [94]. The reason for low strength for DLPs is the distribution of lattice bonds. If one of the important bonds is aligned along the tensile direction or angle rotation is blocked, the Young's modulus will be lower.

Table 6 also presents the anisotropy ratio $A_E = E_{max}/E_{min}$ for all considered phases. Phases with $A_E \geq 2.0$ can be considered as highly anisotropic: CA7, TA3, TA6, LA3, TA8, and LA6. Moreover, for CA7 and LA3, the difference between minimal and maximal Young's modulus is more than 4.0. Such a difference can be explained by the difference in atomic arrangement along different directions. For example, TA6 and LA3 is presented in Figure 1. The smallest anisotropy was found for CA2, CA9, and CB.

Considered DLPs demonstrate a great difference between the values of Poisson's ratio from negative −0.8 to positive 0.66. It can be seen from Figure 2 that for some DLPs, a minimal Poisson's ratio is very close to zero; for example, for CA3, TA1, TA6, and CA2. The near-zero Poisson's ratio indicates that when DLPs are stretched or compressed in one direction, their size in a transverse direction can remain almost unchanged. It opens new opportunities in properties' control via atomic rearrangement. This idea of property control was previously discussed especially for Poisson's ratio [95,96]. The new approach to the search for auxetics has been introduced in [95], which involves simultaneous modification of the microscopic structure of the system and intermolecular interactions. Structure modifications at the microscopic level, which strongly influence Poisson's ratio, were recently discussed in [96,97]. The same approach can be applied to the considered DLPs to control their rate of auxeticity.

Phases CA3, CA7, TA6, LA3, CA2, TB, and LA6 possess a negative Poisson's ratio. This zero in-plane Poisson's ratio, together with the negative out-of-plane Poisson's ratio, shows that the DLPs can be auxetic materials. In accordance with [46], some DLPs are partially auxetic since they possess negative Poisson's ratio in some directions, and some are non-auxetic at all. There are no completely auxetic systems among the considered DLPs. Nevertheless, the class of these DLPs is an excellent example of the realization of different possible behaviors within one system: partially auxetic and non-auxetic structures. The same was previously shown within one system, starting from entirely isotropic elastic properties to strongly anisotropic and showing non-auxetic, partially auxetic, and auxetic behaviors [98]. Moreover, in [99], the methodology for determining the degree of auxeticity was proposed, which allows for analysis of structures with Poisson's ratio from 1 to −1.

Interestingly, tubulane TA6 is a partial auxetic and possesses a considerable difference in Poisson's ratio along different directions [73]. The minimal Poisson's ratio for TA6 is −0.01. For tubulane TA6, the biggest difference between maximal and minimal values was found ($\nu_{max} - \nu_{min} = 0.63$). Averaged over all directions, Poisson's ratio for TA6 is positive and equal to 0.17. For tubulane TA6, maximal Young's modulus is observed in the [100] and [010] directions along which the covalent bonds are aligned. This explains the high elastic modulus.

To understand the structural evolution of DLPs under tension, the in-plane deformation mechanism is investigated based on the geometry analysis. Figure 3 presents stress–strain curves for all tubulanes, fulleranes, and DLPs based on graphene under tension. As can be seen, stress–strain curves for all DLPS are very similar but can be divided into two groups: first group TA1, TA5, LA6, and CA3; second group TA3, TA6, TB, LA3, CA2, and CA8; third group CB, CA9, CA7, and TA8, as shown in Figure 3. Stress–strain curves are shown until the crystal structure is stable. As can be seen, the deformation behavior of all tubulanes is very similar, with an even close limit of the first phase transformation. Such phase transformations are very common for covalent crystals [64]. Despite this, phase transformation is of high interest, and it is not described in the present work since it requires special analysis. For the third group, a sharp increase in the stress on the

last deformation stage is explained by a fast increase in all lattice parameters, while valent angles almost cannot be changed further. There is no clear phase transformation at this stage, just bond elongation. If compared with the values of Young's modulus, it can be seen that phases TA1, TA3, and TA5 possess a lower elastic modulus than TA7, TA8, and TB as well as lower critical strain and stress.

For such carbon structures, the overall deformation under tension is attributed to two deformation mechanisms that are, respectively, elongation and rotation of bonds. Thus, all the deformation mechanisms can be analyzed based on changes in covalent bonds and angles in the structure. The change in deformation mode can be traced by changes in the pressure–strain state, bonds' length–strain state, and value of valent angles–strain state: there is a correlation between all data. A strong correlation between the pressure–strain curve and changes in bond length and values of valent angles is observed. Also, there is a correlation with the stress distribution during tension: if the main contribution is because of the elongation of bonds, normal stress components are dominant; if the main contribution is because of changing valent angles, shear stress components are also meaningful. For structures of group I, the curve slope for $p(\varepsilon)$ is much lower than for structures of group II and III. An analysis of all the lattice parameters $L_1$–$L_4$ or valent angles (the number of independent valent angles is from three to five for different DLPs) cannot be performed in frames of one work. But, the main characteristics can be analyzed. During the elastic region, for all DLPs, deformation is defined by changes in the lattice constants. Commonly, if the bond is aligned with the tensile direction, it decreases the strength of DLP. Contrary, if bonds lie at some angle to the tensile direction, deformation is defined by changes in valent angles which increase the final strength. In region II, a main contribution is made by a change in the lattice parameters (bond length increase for 1–4%). A slight change in the valent angles is found (about 1–2%). On stage II, changes in the lengths of valence bonds is the main deformation mechanism. On the third stage, again, the main mechanism is the changes in valent bonds.

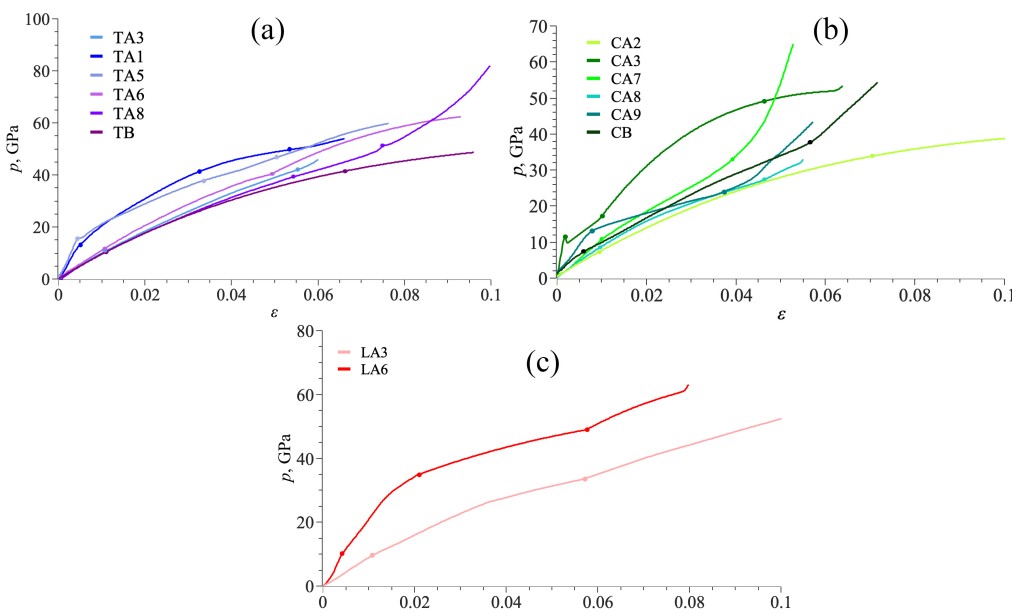

**Figure 3.** Stress–strain curves under tension for tubulanes (**a**), fulleranes (**b**), and DLPs based on graphene (**c**).

For better understanding, Figure 4 shows the stress–strain curve for TA1; as an example, together with the snapshots of the part of the simulation cell. The radial distribution function (RDF) is also presented at critical points. Each color for RDF corresponds to the same color of the star on the stress–strain curve and to the corresponding strain value: orange for $\varepsilon = 0$; blue for $\varepsilon = 0.004$; dark green for $\varepsilon = 0.01$; light green for $\varepsilon = 0.04$; purple

for $\varepsilon = 0.066$; and yellow for $\varepsilon = 0.075$. As seen, RDFs are shown into neighboring points in the stress–strain curve for comparison, but not in one plot for clearance. Until point 1, the elastic deformation took place with a small increase in the bond lengths with almost no changes in valent angles. Interestingly, the third peak moved from 2.3 Å to 2 Å and became two times smaller. Between points 2 and 3, continuous changes in the lattice took place; however, RDFs at two points are very similar. After point 4, irreversible lattice distortion took place: the short-range lattice order is visualized by RDF. However, at this limit, there is no fracture, but a phase transformation with the break of long-order periodicity.

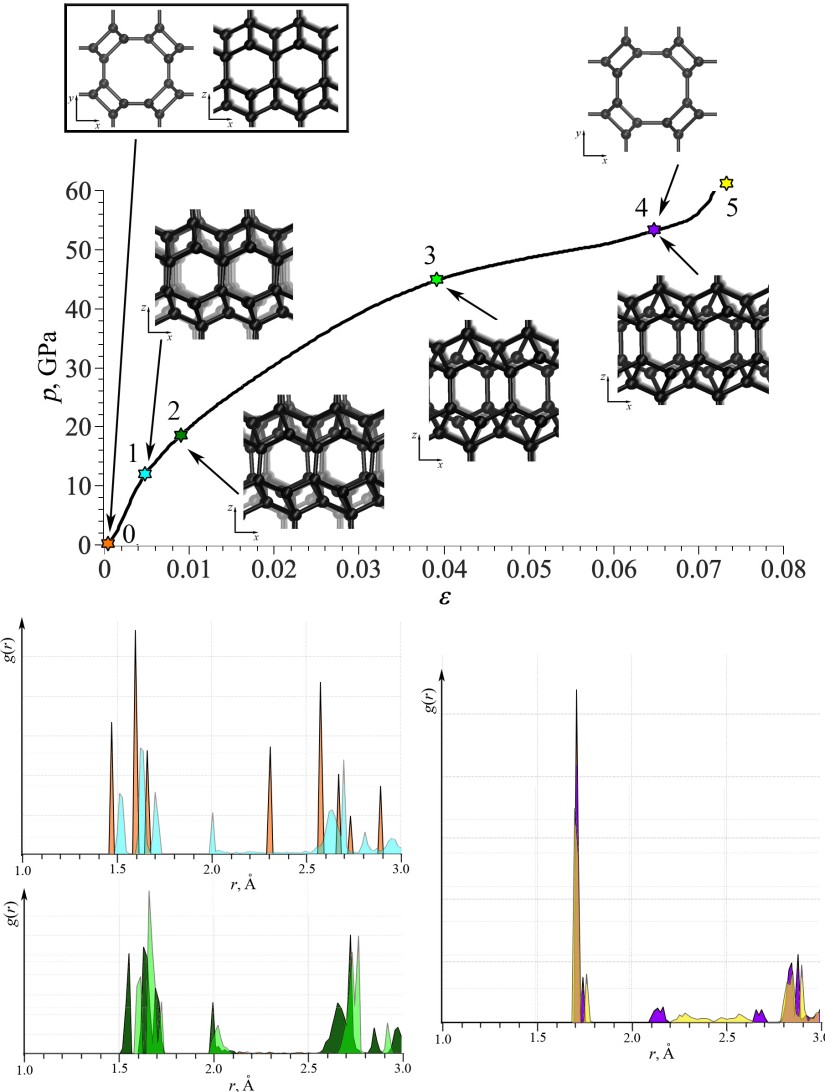

**Figure 4.** Stress–strain curve under tension for TA1. Radial distribution functions at critical points 0–5. Snapshots of the part of TA1 under tension is also presented. Each color for RDF corresponds to the same color of the star on the stress–strain curve and to the corresponding strain value: orange for $\varepsilon = 0$; blue for $\varepsilon = 0.004$; dark green for $\varepsilon = 0.01$; light green for $\varepsilon = 0.04$; purple for $\varepsilon = 0.066$; and yellow for $\varepsilon = 0.075$.

Snapshots of the part of the simulation cell show that the main changes took place in the $xz$ plane, while the structure in $xy$ plane remain almost unchanged (points 0 and 4). Only slight changes in the bond lengths can be seen in the $xy$ projection with a very small rotations, not exceeding 1%. While in the $xz$ plane, deformation is very complex, including changes in the valent and dihedral angles. Shear mode with the mutual shift of crystallographic planes took place. The same changes can be found for the $yz$ plane.

For such DLP structures, deformation mechanisms considerably depends on the number of effective lattice parameters and important angles in the lattice.

## 4. Conclusions

A detailed study has been performed to investigate the mechanical properties of diamond-like phases of different morphology using MD simulations. The deformation mechanisms during tensile loading of diamond-like phases were analyzed based on changes in covalent bonds and valent angles. The considered diamond-like phases are a special class of structures based especially on three carbon polymorphs.

Based on the stiffness coefficients obtained from molecular dynamics, elastic constants (Young's modulus and Poisson's ratio) of diamond-like phases were calculated. The maximum Young's modulus for TA6, TB, LA3, and LA6 was found to be greater than 1 TPa, i.e., greater than the maximum value for diamond and graphene. Most of the structures demonstrate high anisotropy of Young's modulus and Poisson's ratio. Half of the stable DLPs (CA3, CA7, TA6, LA3, CA2, TB, and LA6) are partial auxetics. Interestingly, three DLPs possess a Poisson's ratio close to zero (TA6, CA2, TA1).

During tension, several different deformation stages are revealed, including elastic and inelastic deformation, characterized by different tensile mechanisms. Tensile deformation for DLPs occurs due to the continuous change in covalent bonds or valence angles. All DLPs differ in the number of basic lattice parameters and basic valence angles, and thus, the particular deformation mechanisms are different: for some structures, tension occurs mainly due to a change in the lattice parameters, and for others, it is due to the changes in valence angles or a simultaneous change in bonds and angles. Our results suggest that diamond-like phases have extraordinary mechanical properties, making them good materials for protective coatings.

**Funding:** The research is funded by the Ministry of Science and Higher Education of the Russian Federation as part of the World-class Research Center program: Advanced Digital Technologies (contract No. 075-15-2022-311 dated 20 April 2022).

**Data Availability Statement:** Data are contained within the article.

**Conflicts of Interest:** The author declares no conflicts of interest.

## Abbreviations

The following abbreviations are used in this manuscript:

| | |
|---|---|
| DLPs | diamond-like phases |
| MD | molecular dynamics |
| CNT | carbon nanotubes |
| RDF | radial distribution function |

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
