# Peer review of "An Overview of Mechanical Properties of Diamond-like Phases under Tension"

_nanomaterials, doi:10.3390/nano14020129_

Round 1
Reviewer 1 Report
Comments and Suggestions for Authors
The work concerns the elastic properties of diamond-like phases. The results regarding the auxetic properties and hardness of some of the tested structures are particularly interesting. The subject of the work is current and interesting. However, the reviewed manuscript lacks a lot of basic information that could introduce the reader to the research topic (feeble introduction). Moreover, it also lacks details of computer simulations (this applies to both the methodology and the studied systems) making it impossible to repeat this research. In my opinion, the most interesting part of the work concerns auxetic properties but even the definition of auxetics is missing. The discussion of the results is superficial - there is a lack of quantitative assessment of the results obtained. For example, there is a lack of analysis of changes occurring in the parameters of unit cells. Moreover, the paper does not provide the parameters of unit cells, the sizes of the studied systems, or even the figures of all studied systems, etc. An analysis only for one system (the research concerns 14 systems belonging to different symmetry groups) based on a visual assessment of structure changes in the snapshots is unacceptable. The conclusions are rather weakly supported by the results contained in this manuscript. For example, in the section Conclusion one can find the following sentences: “The deformation mechanisms during tensile loading of diamond-like phases were analyzed based on changes in covalent bonds and valent angles.”, and “All DLPs differ in the number of basic lattice parameters and basic valence angles, and thus, the particular deformation mechanisms are different: for some structures, tension occurs mainly due to a change of the lattice parameters, for others – due to the changes in valence angles, or a simultaneous change of bonds and angles.” Unfortunately, the information based on which such conclusions were drawn is simply not available in this manuscript!
The paper can be considered for publication after major revision. In particular, it would be important if the Author addressed the issues listed below in the manuscript.
1) The Introduction deserves special attention and should be significantly improved. It is necessary to amend and supplement the Introduction. In the present Introduction, one can find mainly a quite random and chaotic description of some diamond-like phases. But there is a complete lack of any introduction and references to auxetic and partially auxetic systems.
2) It would be advisable to introduce the reader to auxetic topics in the introduction. The literature on auxetics is very extensive. However, a discussion of some fundamental works in this field, and not only those recently published, will be a valuable introduction to the topic. Here one should suggest a discussion in the manuscript of the below works:
a) One of the first papers on mechanical model with negative Poisson’s ratio (auxetic) was published by Almgren: [AN ISOTROPIC 3-DIMENSIONAL STRUCTURE WITH POISSON RATIO=-1; By: ALMGREN, RF; JOURNAL OF ELASTICITY, Volume: 15, Issue: 4, Pages: 427-430, Published: 1985],
b) The first auxetic foam was manufactured by Lakes: [FOAM STRUCTURES WITH A NEGATIVE POISSONS RATIO; By: LAKES, R; SCIENCE Volume: 235 Issue: 4792 Pages: 1038-1040 Published: FEB 27 1987],
c) The first microscopic model (of the hard cyclic hexamers) with a thermodynamically stable isotropic phase exhibiting negative Poisson’s ratio was simulated by Monte Carlo method in [CONSTANT THERMODYNAMIC TENSION MONTE-CARLO STUDIES OF ELASTIC PROPERTIES OF A TWO-DIMENSIONAL SYSTEM OF HARD CYCLIC HEXAMERS; By: WOJCIECHOWSKI, KW; Molecular Physics, Volume 61, Issue: 5, Pages: 1247-1258, Published: 1987] and then its certain generalization (soft cyclic hexamers) was analytically solved in the static limit (at zero temperature) in: [TWO-DIMENSIONAL ISOTROPIC SYSTEM WITH A NEGATIVE POISSON RATIO; By: WOJCIECHOWSKI, KW; PHYSICS LETTERS A, Volume: 137, Issue: 1-2, Pages: 60-64, Published: MAY 1 1989],
d) The term "auxetic" was introduced in [AUXETIC POLYMERS - A NEW RANGE OF MATERIALS; By: K. E. Evans, ENDEAVOUR, Volume: 15, Issue: 4, Pages: 170–174, Published: 1991],
e) A pioneering work on composite auxetics is [COMPOSITE-MATERIALS WITH POISSON RATIOS CLOSE TO -1; By: MILTON, GW; JOURNAL OF THE MECHANICS AND PHYSICS OF SOLIDS Volume: 40 Issue: 5 Pages: 1105-1137 Published: JUL 1992],
f) A very important paper on cubic auxetics is [Negative Poisson's ratios as a common feature of cubic metals; By: Baughman, RH; Shacklette, JM; Zakhidov, AA; et al.; NATURE Volume: 392 Issue: 6674 Pages: 362-365 Published: MAR 26 1998],
g) A well-known rotational planar model was proposed in [Auxetic behavior from rotating squares; By: Grima, JN; Evans, KE; JOURNAL OF MATERIALS SCIENCE LETTERS Volume: 19 Issue: 17 Pages: 1563-1565 Published: SEP 2000],
h) Computer simulations of an important molecular auxetic were described in [Mechanism for negative Poisson ratios over the alpha-beta transition of cristobalite, SiO2: A molecular-dynamics study; By: Kimizuka, H; Kaburaki, H; Kogure, Y; PHYSICAL REVIEW LETTERS Volume: 84 Issue: 24 Pages: 5548-5551 Published: JUN 12 2000].
It would also be worth to mention more recent theoretical or simulation works concerning some fundamental auxetic structures and mechanisms at the micro level, related to this paper, for example:
i) Auxeticity of graphene was for the first time simulated in: [Tailoring Graphene to Achieve Negative Poisson's Ratio Properties, by Grima, Joseph N.; Winczewski, Szymon; Mizzi, Luke; et al., ADVANCED MATERIALS Volume: 27 Issue: 8 Pages: 1455-+ Published: FEB 25 2015],
j) A recent review on auxetic solids was published in [Negative-Poisson's-Ratio Materials: Auxetic Solids; By: Lakes, Roderic S.; ANNUAL REVIEW OF MATERIALS RESEARCH, VOL 47 Book Series: Annual Review of Materials Research Volume: 47 Pages: 63-81 Published: 2017],
3) In the context of the discussion about the anisotropy of elastic properties, mention should be made of the classification of auxetic materials generally accepted in the literature. Namely, "auxetic" (or completely auxetic) for systems with negative Poisson's ratio in all directions, "partially auxetic" for systems with negative Poisson's ratio in some directions, and "non-auxetic" for the rest cases have to be used. This can be found in [Branka, AC; Heyes, DM; Wojciechowski, KW; PHYSICA STATUS SOLIDI B-BASIC SOLID STATE PHYSICS, Volume: 248, Issue: 1, Pages: 96-104, Published: SEP 2011].
4) Is there a (completely) auxetic system among the studied systems? The Author mentioned that the studied system may have partially auxetic and non-auxetic properties. In this context, there is an excellent example of the realization of all possible behaviors within one system, starting from entirely isotropic elastic properties to strongly anisotropic and showing non-auxetic, partially auxetic, and auxetic behaviors: [Auxetic, Partially Auxetic, and Nonauxetic Behaviour in 2D Crystals of Hard Cyclic Tetramers, Tretiakov, K. V.; Wojciechowski, K. W. PHYSICA STATUS SOLIDI-RAPID RESEARCH LETTERS Volume: 14 Issue: 7 Article Number: 2000198 Published: JUL 2020].
5) Is it possible to relate the studied systems to an ideal auxetic, i.e. the one with Poisson’s ratio equal to -1? It can be done by determining the degree of auxeticity of the studied systems, which has been proposed by Piglowski, P.M. et al.; SOFT MATTER Volume: 13, Issue: 43, Pages: 7921-7916, Published: NOV 21 2017.
6) The reviewed work presents the results of computer simulations, where changes in the microstructure of the system and their consequences on the elastic properties are considered. Recently, a new approach to the search for auxetics has been introduced, which involves simultaneous modification of the microscopic structure of the system and intermolecular interactions:
a) Composite structures on a molecular level: [Partial auxeticity induced by nanoslits in the Yukawa crystal By: Piglowski, Pawel M.; et al.; PHYSICA STATUS SOLIDI-RAPID RESEARCH LETTERS Volume: 10 Issue: 7 Pages: 566-569 Published: JUL 2016]
b) Structure modifications at the microscopic level, which strongly influence Poisson’s ratio, were recently discussed in: [Auxetic Properties of a f.c.c. Crystal of Hard Spheres with an Array of [001]-Nanochannels Filled by Hard Spheres of Another Diameter; By: Narojczyk, J. W.; et al.; PHYSICA STATUS SOLIDI B-BASIC SOLID STATE PHYSICS Volume: 256 Article Number: 1800611 Published: JAN 2019] and in [Removing Auxetic Properties in f.c.c. Hard Sphere Crystals by Orthogonal Nanochannels with Hard Spheres of Another Diameter; By: Narojczyk, Jakub W.; et al.; MATERIALS; Volume: 15 Article Number: 1134 Published: FEB 2022; DOI: 10.3390/ma15031134].
7) In the reviewed work, the Author places particular emphasis on the hardness of the studied systems. The most recent work (Phys. Rev. E 108, 045003 (2023)) on this topic concerns of hardening of fcc hard-sphere crystals by structure modification that complements the present work.
8) Figures and the unit cell parameters of all structures should be provided (either in the main text or in supplementary information).
9) Quantitative analysis of unit cell parameters of all structures examined should be given.
10) The manuscript should include all details necessary to replicate the research results. This applies to both the tested systems and the details of the methods used (system sizes, statistical ensemble, length of simulations, time of equilibration of samples, etc.). Thermostat and barostat coupling constants should be given.
11) The method for determining the elastic constants should be described separately in great detail.
12) The measurement (statistical) errors in all presented results are absent. It should be corrected.
13) At what temperatures were the simulations performed?
14) How does the density of the systems change?
15) In the discussion, the Author mentions "phase transformation". It should be explained what kind of "transformation" are these, and what are they related to?
16) In general, elastic properties are described using a 4th-order tensor; the manuscript uses a matrix description, probably using the Voigt notation. Is that so? If this is the case, the Author need to provide formulas for transitioning from one notation to another.
17) In the introduction, the Author mentioned about analysis of "the compliance constants". In the section Simulation Details, one can find "calculate stiffness constants" and then further "Stiffness coefficients" and "Stiffness constants" (Table 1). It would be good to specify precisely what the Author is calculating – formulas used in the calculation should be provided.
18) Figure 3 shows the results for tubulanes. What does it look like for other systems?
19) In Figure 3, not all curves start from zero. Might the Author explain why is it like this?
20) Figure 4 is not sufficiently well described. There are no colors explained in the drawings.
21) The author mentions in the manuscript the Born stability criteria and that the tested systems meet these criteria. These important results regarding stability criteria should be presented in the paper.
22) In the context of the studied structures' anisotropy, it is unclear in which directions the Poisson's ratio values are given.
23) Were simulations and comparisons made for other interatomic potentials than AIREBO? To what extent do the simulation results depend on the choice of interatomic potential?
24) On Page 7, one can find "The number of stages of non-elastic deformation depends on the DLP morphology and can be equal to 3–4." Based on what criteria were these zones designated?
25) All references should be carefully checked - the reference [13] is empty.
Author Response
Dear Editor and Reviewers:
I sincerely thank Editor and Reviewers for thoroughly examining our manuscript entitled “An overview of deformation mechanisms of diamond-like phases under tension” and providing very constructive comments to guide our revision. I have carefully revised the manuscript in accordance with the Reviewers comments. I responded to the comments point by point and highlighted the changes in the revised manuscript in red color.
Reviewer #1:
The results regarding the auxetic properties and hardness of some of the tested structures are particularly interesting. The subject of the work is current and interesting. However, the reviewed manuscript lacks a lot of basic information that could introduce the reader to the research topic (feeble introduction). Moreover, it also lacks details of computer simulations (this applies to both the methodology and the studied systems) making it impossible to repeat this research. The discussion of the results is superficial - there is a lack of quantitative assessment of the results obtained. The conclusions are rather weakly supported by the results contained in this manuscript. The paper can be considered for publication after major revision.
Reply(R): We thank the reviewer for such a careful reading. The lack of the mentioned information is explained by our attempt to write a short letter to eliminate the main mechanical characteristics. We agree that in this case, the lack of additional descriptions decreases the readability and soundness of the manuscript. We have tried to answer all the mentioned issues and improve our manuscript.
Q1: The Introduction deserves special attention and should be significantly improved. It is necessary to amend and supplement the Introduction. In the present Introduction, one can find mainly a quite random and chaotic description of some diamond-like phases. But there is a complete lack of any introduction and references to auxetic and partially auxetic systems.
R1: I appreciate the comment. In the Introduction part I decided to describe different new DLPs and their common mechanical properties and applications. I totally agree that the description of auxetics is very important. The introduction was considerably improved. All the issues mentioned by the Reviewer are described in the revised version.
Q2: It would be advisable to introduce the reader to auxetic topics in the introduction. The literature on auxetics is very extensive. However, a discussion of some fundamental works in this field, and not only those recently published, will be a valuable introduction to the topic. Here one should suggest a discussion in the manuscript of the below works:
R2: I appreciate the comment. All the works mentioned by the Reviewer are added and described in the Introduction.
- a) One of the first papers on mechanical model with negative Poisson’s ratio (auxetic) was published by Almgren: [AN ISOTROPIC 3-DIMENSIONAL STRUCTURE WITH POISSON RATIO=-1; By: ALMGREN, RF; JOURNAL OF ELASTICITY, Volume: 15, Issue: 4, Pages: 427-430, Published: 1985],
- b) The first auxetic foam was manufactured by Lakes: [FOAM STRUCTURES WITH A NEGATIVE POISSONS RATIO; By: LAKES, R; SCIENCE Volume: 235 Issue: 4792 Pages: 1038-1040 Published: FEB 27 1987],
- c) The first microscopic model (of the hard cyclic hexamers) with a thermodynamically stable isotropic phase exhibiting negative Poisson’s ratio was simulated by Monte Carlo method in [CONSTANT THERMODYNAMIC TENSION MONTE-CARLO STUDIES OF ELASTIC PROPERTIES OF A TWO-DIMENSIONAL SYSTEM OF HARD CYCLIC HEXAMERS; By: WOJCIECHOWSKI, KW; MOLECULAR PHYSICS, Volume 61, Issue: 5, Pages: 1247-1258, Published: 1987] and then its certain generalization (soft cyclic hexamers) was analytically solved in the static limit (at zero temperature) in: [TWO-DIMENSIONAL ISOTROPIC SYSTEM WITH A NEGATIVE POISSON RATIO; By: WOJCIECHOWSKI, KW; PHYSICS LETTERS A, Volume: 137, Issue: 1-2, Pages: 60-64, Published: MAY 1 1989],
- d) The term "auxetic" was introduced in [AUXETIC POLYMERS - A NEW RANGE OF MATERIALS; By: K. E. Evans, ENDEAVOUR, Volume: 15, Issue: 4, Pages: 170–174, Published: 1991],
- e) A pioneering work on composite auxetics is [COMPOSITE-MATERIALS WITH POISSON RATIOS CLOSE TO -1; By: MILTON, GW; JOURNAL OF THE MECHANICS AND PHYSICS OF SOLIDS Volume: 40 Issue: 5 Pages: 1105-1137 Published: JUL 1992],
- f) A very important paper on cubic auxetics is [Negative Poisson's ratios as a common feature of cubic metals; By: Baughman, RH; Shacklette, JM; Zakhidov, AA; et al.; NATURE Volume: 392 Issue: 6674 Pages: 362-365 Published: MAR 26 1998],
- g) A well-known rotational planar model was proposed in [Auxetic behavior from rotating squares; By: Grima, JN; Evans, KE; JOURNAL OF MATERIALS SCIENCE LETTERS Volume: 19 Issue: 17 Pages: 1563-1565 Published: SEP 2000],
- h) Computer simulations of an important molecular auxetic were described in [Mechanism for negative Poisson ratios over the alpha-beta transition of cristobalite, SiO2: A molecular-dynamics study; By: Kimizuka, H; Kaburaki, H; Kogure, Y; PHYSICAL REVIEW LETTERS Volume: 84 Issue: 24 Pages: 5548-5551 Published: JUN 12 2000].
It would also be worth to mention more recent theoretical or simulation works concerning some fundamental auxetic structures and mechanisms at the micro level, related to this paper, for example:
- i) Auxeticity of graphene was for the first time simulated in: [Tailoring Graphene to Achieve Negative Poisson's Ratio Properties, by Grima, Joseph N.; Winczewski, Szymon; Mizzi, Luke; et al., ADVANCED MATERIALS Volume: 27 Issue: 8 Pages: 1455-+ Published: FEB 25 2015],
- j) A recent review on auxetic solids was published in [Negative-Poisson's-Ratio Materials: Auxetic Solids; By: Lakes, Roderic S.; ANNUAL REVIEW OF MATERIALS RESEARCH, VOL 47 Book Series: Annual Review of Materials Research Volume: 47 Pages: 63-81 Published: 2017],
Q3: In the context of the discussion about the anisotropy of elastic properties, mention should be made of the classification of auxetic materials generally accepted in the literature. Namely, "auxetic" (or completely auxetic) for systems with negative Poisson's ratio in all directions, "partially auxetic" for systems with negative Poisson's ratio in some directions, and "non-auxetic" for the rest cases have to be used. This can be found in [Branka, AC; Heyes, DM; Wojciechowski, KW; PHYSICA STATUS SOLIDI B-BASIC SOLID STATE PHYSICS, Volume: 248, Issue: 1, Pages: 96-104, Published: SEP 2011].
R3: I appreciate the comment. Corresponding information and discussion were added.
Q4: Is there a (completely) auxetic system among the studied systems? The Author mentioned that the studied system may have partially auxetic and non-auxetic properties. In this context, there is an excellent example of the realization of all possible behaviors within one system, starting from entirely isotropic elastic properties to strongly anisotropic and showing non-auxetic, partially auxetic, and auxetic behaviors: [Auxetic, Partially Auxetic, and Nonauxetic Behaviour in 2D Crystals of Hard Cyclic Tetramers, Tretiakov, K. V.; Wojciechowski, K. W. PHYSICA STATUS SOLIDI-RAPID RESEARCH LETTERS Volume: 14 Issue: 7 Article Number: 2000198 Published: JUL 2020].
R4: I appreciate the comment. There are no completely auxetic systems among these DLPs. But the idea that “there is an excellent example of the realization of all possible behaviors within one system” is very interesting and definitely should be described in the manuscript. I have added corresponding information and references.
Q5: Is it possible to relate the studied systems to an ideal auxetic, i.e. the one with Poisson’s ratio equal to -1? It can be done by determining the degree of auxeticity of the studied systems, which has been proposed by Piglowski, P.M. et al.; SOFT MATTER Volume: 13, Issue: 43, Pages: 7921-7916, Published: NOV 21 2017.
R5: I appreciate the comment. It was discussed in accordance with the mentioned work.
Q6: The reviewed work presents the results of computer simulations, where changes in the microstructure of the system and their consequences on the elastic properties are considered. Recently, a new approach to the search for auxetics has been introduced, which involves simultaneous modification of the microscopic structure of the system and intermolecular interactions:
- a) Composite structures on a molecular level: [Partial auxeticity induced by nanoslits in the Yukawa crystal By: Piglowski, Pawel M.; et al.; PHYSICA STATUS SOLIDI-RAPID RESEARCH LETTERS Volume: 10 Issue: 7 Pages: 566-569 Published: JUL 2016]
- b) Structure modifications at the microscopic level, which strongly influence Poisson’s ratio, were recently discussed in: [Auxetic Properties of a f.c.c. Crystal of Hard Spheres with an Array of [001]-Nanochannels Filled by Hard Spheres of Another Diameter; By: Narojczyk, J. W.; et al.; PHYSICA STATUS SOLIDI B-BASIC SOLID STATE PHYSICS Volume: 256 Article Number: 1800611 Published: JAN 2019] and in [Removing Auxetic Properties in f.c.c. Hard Sphere Crystals by Orthogonal Nanochannels with Hard Spheres of Another Diameter; By: Narojczyk, Jakub W.; et al.; MATERIALS; Volume: 15 Article Number: 1134 Published: FEB 2022; DOI: 10.3390/ma15031134].
R6: I appreciate the comment. Additional discussion is added in accordance with the reviewer advice.
Q7: In the reviewed work, the Author places particular emphasis on the hardness of the studied systems. The most recent work (Phys. Rev. E 108, 045003 (2023)) on this topic concerns of hardening of fcc hard-sphere crystals by structure modification that complements the present work.
R7: I appreciate the comment. Hardness of the structures was discussed in accordance with the mentioned work.
Q8: Figures and the unit cell parameters of all structures should be provided (either in the main text or in supplementary information).
R8: I appreciate the comment. Figure 1 was changed to show all the considered phases. I did not present all of them just to shorten the manuscript. But I agree that it is very important. The new tables 1-3 were also added with the parameters of all DLPs.
Q9: Quantitative analysis of unit cell parameters of all structures examined should be given.
R9: I appreciate the comment. In fact, it is very complicated to describe all the structural changes for 14 DLPs. For each DLP, a number of lattice parameters is 4, and a number of independent valent angles is from 3 to 5 for different DLPs. Analysis showed that qualitative changes are very similar for different phases. Partially some results are presented in [72,73,87,88], and here I tried to present the overview of mechanical properties to show the perspectives of further application of DLPs. I additionally describe all possible changes, show all the stress-strain curves, and analyze different deformation stages. I also add quantitative characteristics of structural changes. Additional analysis of the unit cell parameters was added.
Q10: The manuscript should include all details necessary to replicate the research results. This applies to both the tested systems and the details of the methods used (system sizes, statistical ensemble, length of simulations, time of equilibration of samples, etc.). Thermostat and barostat coupling constants should be given.
R10: I appreciate the comment. All the required information was added.
Q11: The method for determining the elastic constants should be described separately in great detail.
R11: I appreciate the comment. The method for determining the elastic constants was described in details.
Q12: The measurement (statistical) errors in all presented results are absent. It should be corrected.
R12: I appreciate the comment. The measurement errors were added to Tab. 4 and Tab.5.
Q13: At what temperatures were the simulations performed?
R13: I appreciate the comment. Calculation of elastic constants were performed at 0 K. Calculations of tensile strain was performed at 0 K. Additional information was added to the manuscript.
Q14: How does the density of the systems change?
R14: I appreciate the comment. Tension is conducted in such a way to preserve the simulation cell volume. Thus, the density of the system was not changed. Additional information was added to the manuscript.
Q15: In the discussion, the Author mentions "phase transformation". It should be explained what kind of "transformation" are these, and what are they related to?
R15: I appreciate the comment. However, stress-strain curves are presented until phase transformation happens. The point is that this study requires special attention and cannot be fully described in the frames of the present work. Additional explanations were added.
Q16: In general, elastic properties are described using a 4th-order tensor; the manuscript uses a matrix description, probably using the Voigt notation. Is that so? If this is the case, the Author need to provide formulas for transitioning from one notation to another.
R16: I appreciate the comment. The reviewer is totally right. In the present work, I used the Voigt notation. However, I did not describe this because of the limitation of the paper length. Also, I just use all well-known formulas, which were previously published in [72,73,87,88]. Thus, I added an additional explanation on this with all the required references.
Q17: In the introduction, the Author mentioned about analysis of "the compliance constants". In the section Simulation Details, one can find "calculate stiffness constants" and then further "Stiffness coefficients" and "Stiffness constants" (Table 1). It would be good to specify precisely what the Author is calculating – formulas used in the calculation should be provided.
R17: I thank the reviewer for the comment. Formulas for the compliance and stiffness constants were provided in the manuscript.
Q18: Figure 3 shows the results for tubulanes. What does it look like for other systems?
R18: I appreciate the comment. The stress-strain curves for fulleranes and DLPs based on graphene were added to Fig. 3. The description of the difference between different phases was added.
Q19: In Figure 3, not all curves start from zero. Might the Author explain why is it like this?
R19: I thank the referee for such a careful reading. It was my mistake. Occasionally I did not relax these structures (for which stress-strain curves are not from zero) to the global energy minimum. I did additional relaxation. Now all the structures loaded after a fully relaxed state and stress-strain curves start from zero.
Q20: Figure 4 is not sufficiently well described. There are no colors explained in the drawings.
R20: I thank the referee for such a careful reading. Additional description of Fig. 4 was added.
Q21: The author mentions in the manuscript the Born stability criteria and that the tested systems meet these criteria. These important results regarding stability criteria should be presented in the paper.
R21: I appreciate the comment. Additional information was presented.
Q22: In the context of the studied structures' anisotropy, it is unclear in which directions the Poisson's ratio values are given.
R22: I appreciate the comment. Figure 2 presents the lowest value of Poisson’s ratio found for this DLP. I wish to show the minimal value to demonstrate the possibility of auxeticity for DLPs. Additional explanations are added.
Q23: Were simulations and comparisons made for other interatomic potentials than AIREBO? To what extent do the simulation results depend on the choice of interatomic potential?
R23: I appreciate the comment. The choice of the interatomic potential is very important. Additional description was added on the applicability of other interatomic potentials.
Q24: On Page 7, one can find "The number of stages of non-elastic deformation depends on the DLP morphology and can be equal to 3–4." Based on what criteria were these zones designated?
R24: I appreciate the comment. Fig. 3 was changed and a corresponding new description of the pressure-strain curves was added. Different deformation stages were additionally described and shown on the curves by colored dots.
Q25: All references should be carefully checked - the reference [13] is empty.
R25: I thank the reviewer for such a careful reading. Ref. [13] is a reference to web site and occasionally was missed. Refs. were checked and revised.
Reviewer 2 Report
Comments and Suggestions for Authors
The theoretical investigation by J.A. Baimova aims to compare DLPs and their mechanical properties, and does so in case of 14 DLPs.
The introduction could benefit from more details regarding the output of the calculations.
References 13 and 43 are both missing.
Auxetic materials mentioned should be further detailed and explained for the broad readership of Nanomaterials.
A detailed comparison is due for TA6’s case, where Poisson’s ratio is close to zero, but Young’s modulus is maximum.
Why are CA2, CA9 and CB among the weakest DLP?
What are the practical implications of this study? What are the envisioned methods by which DLP can be created?
A thorough grammar check should be performed to eliminate typos and language errors. See for instance, “zero Yuong’s modulus” (page 4)
Comments on the Quality of English LanguageMinor editiong required, there are instances where typos and grammar issues can be detected.
Author Response
Dear Editor and Reviewers:
I sincerely thank Editor and Reviewers for thoroughly examining our manuscript entitled “An overview of deformation mechanisms of diamond-like phases under tension” and providing very constructive comments to guide our revision. I have carefully revised the manuscript in accordance with the Reviewers comments. I responded to the comments point by point and highlighted the changes in the revised manuscript in red color.
Q1: The introduction could benefit from more details regarding the output of the calculations.
Reply1 (R1): I appreciate the comment. In the Introduction part I decided to describe different new DLPs and their common mechanical properties and application in short. I totally agree that an additional description of the output of the calculations is required. The introduction was considerably improved.
Q2: References 13 and 43 are both missing.
R2: I thank the reviewer for such a careful reading. Refs. [13,43] are the references to web site and occasionally were missed. Refs. were checked and revised.
Q3: Auxetic materials mentioned should be further detailed and explained for the broad readership of Nanomaterials.
R3: I appreciate the comment. I totally agree that the description of auxetics is very important. The introduction was considerably improved.
Q4: A detailed comparison is due for TA6’s case, where Poisson’s ratio is close to zero, but Young’s modulus is maximum.
R4: I appreciate the comment. In fact, tubulane TA6 is a partial auxetic (crystal that demonstrates both negative and positive values of Poisson's ratio depending on the considered direction of uniaxial strain and lateral direction. Only the minimal value of Poisson’s ratio is presented, while there is a considerable difference in these values. For tubulane TA6, maximal Young’s modulus is observed in [100] and [010] directions along which the covalent bonds are aligned. This explains the high elastic modulus. Additional discussion of the results for TA6 was added.
Q5: Why are CA2, CA9 and CB among the weakest DLP?
R5: I appreciate the comment. The reason for the low strength of DLPs is the distribution of lattice bonds. If one of the important bonds is aligned along a tensile direction or angle rotation is blocked somehow, the structure will show a low Young’s modulus. Additional discussion of the results for CA2, CA9, and CB was added.
Q6: What are the practical implications of this study? What are the envisioned methods by which DLP can be created?
R6: I appreciate the comment. Part of the information was added to the Introduction part.
Q7: A thorough grammar check should be performed to eliminate typos and language errors. See for instance, “zero Yuong’s modulus” (page 4)
R7: I thank the reviewer for such a careful reading. Spelling was carefully checked and misprints were fixed.
Reviewer 3 Report
Comments and Suggestions for Authors
In this work, the author detailed studied the mechanical properties of stable diamond-like phases with different morphology based on fullerenes, carbon nanotubes and graphene by molecular dynamics simulation and calculated the compliance constants, Young’s modulus, and Poisson’s ratio. In addition, they analyzed the deformation mechanism of diamond-like phases based on changes in covalent bonds and valent angles systematically. The results confirm that diamond-like phases possess extraordinary mechanical properties that permit them optional materials for protective coatings. I would recommend the paper for publication after the minor issue that page numbers are missing in some references is addressed.
Author Response
Dear Editor and Reviewers:
I sincerely thank Editor and Reviewers for thoroughly examining our manuscript entitled “An overview of deformation mechanisms of diamond-like phases under tension” and providing very constructive comments to guide our revision. I have carefully revised the manuscript in accordance with the Reviewers comments. I responded to the comments point by point and highlighted the changes in the revised manuscript in red color.
Reviewer:
I would recommend the paper for publication after the minor issue that page numbers are missing in some references is addressed.
Reply: I thank the Reviewer for such a positive estimation. I have tried to correct the references.
Reviewer 4 Report
Comments and Suggestions for Authors
Your paper seems to be introducing characteristics of diamond-like phases and investigating mechanical properties of DLPs based on various carbon allotropes with molecular dynamics simulation. This paper looks well-organized and very interesting. However, some questions arose, so I am leaving the following comments.
1. Based on your paper, DLP is recognized as protective coating. So, hardness of DLP seems to be very important too. Can you describe me why you handled only Young’s modulus and Poisson’s ratio in this paper?
2. It is predicted that mechanical properties of fabricated DLP films should be affected by thickness. Have you conducted or referenced any previous researches on the thickness dependence of the mechanical properties of DLP films?
3. It is very interesting to confirm the phase transformation by comparing the stress-strain curve and the radial distribution function (RDF). I think it would be clearer if it was specified which phase the part of TA1 sample transformed into and what the proportion of the transformed phase was.
4. Temperature and pressure in the simulation space are also likely to influence the mechanical behaviors under tension. What are the temperature and pressure conditions?
Comments on the Quality of English LanguageMinor editing of English language required
Author Response
Dear Editor and Reviewers:
I sincerely thank Editor and Reviewers for thoroughly examining our manuscript entitled “An overview of deformation mechanisms of diamond-like phases under tension” and providing very constructive comments to guide our revision. I have carefully revised the manuscript in accordance with the Reviewers comments. I responded to the comments point by point and highlighted the changes in the revised manuscript in red color.
Q1: Based on your paper, DLP is recognized as protective coating. So, hardness of DLP seems to be very important too. Can you describe me why you handled only Young’s modulus and Poisson’s ratio in this paper?
Reply1 (R1): I appreciate the comment. The hardness of the structures was estimated in other works. I have added some references to the Introduction part.
Q2: It is predicted that mechanical properties of fabricated DLP films should be affected by thickness. Have you conducted or referenced any previous researches on the thickness dependence of the mechanical properties of DLP films?
R2: I appreciate the comment. In fact, we have considered infinite crystals due to the periodicity of the simulation cell. Definitely, this issue is very interesting and it is one of my plans to consider the effect of thickness on the mechanical behavior of DLP as a coating. However, to date, I have not conducted such calculations. However, I have added additional information about this issue.
Q3: It is very interesting to confirm the phase transformation by comparing the stress-strain curve and the radial distribution function (RDF). I think it would be clearer if it was specified which phase the part of TA1 sample transformed into and what the proportion of the transformed phase was.
R3: I appreciate the comment. This phase transformation is very important and interesting; however, stress-strain curves are presented until phase transformation happens. The point is that this study requires special attention and cannot be fully described in the frames of the present work. Additional explanations were added.
Q4: Temperature and pressure in the simulation space are also likely to influence the mechanical behaviors under tension. What are the temperature and pressure conditions?
R4: I appreciate the comment. Calculation of elastic constants was performed at 0 K. Calculations of tensile strain were performed at 0 K. However, I also checked tensile deformation at 300 K. Temperature decreased fracture strain and stress, but only slightly. Additional information was added to the manuscript.
Round 2
Reviewer 1 Report
Comments and Suggestions for Authors
The author responded to all my comments. The work can be published after correcting an inaccuracy in the description of Fig. 3. There is a lack of "(c)". Instead " ... fulleranes (b) and DLPs based on graphene." should be " ... fulleranes (b), and DLPs based on graphene (c)."
Author Response
I thank the Reviewer for his great efforts and valuable comments. I have corrected the discription of Fig. 3.